

# The soil microbiomics of intact, degraded and partially-restored semi-arid succulent thicket (Albany Subtropical Thicket)

Micaela Schagen[1],*, Jason Bosch[1],*, Jenny Johnson[1], Robbert Duker[2], Pedro Lebre[1], Alastair J. Potts[2] and Don A. Cowan[1]

[1] Centre for Microbial Ecology and Genomics, Department of Biochemistry, Genetics and Microbiology, University of Pretoria, Pretoria, Gauteng, South Africa
[2] Botany Department, South Campus, Nelson Mandela University, Port Elizabeth, Eastern Cape, South Africa
* These authors contributed equally to this work.

Corresponding author
Don A. Cowan, don.cowan@up.ac.za

## ABSTRACT

This study examines the soil bacterial diversity in the *Portulacaria afra*-dominated succulent thicket vegetation of the Albany Subtropical Thicket biome; this biome is endemic to South Africa. The aim of the study was to compare the soil microbiomes between intact and degraded zones in the succulent thicket and identify environmental factors which could explain the community compositions. Bacterial diversity, using 16S amplicon sequencing, and soil physicochemistry were compared across three zones: intact (undisturbed and vegetated), degraded (near complete removal of vegetation due to browsing) and restored (a previously degraded area which was replanted approximately 11 years before sampling). Amplicon Sequence Variant (ASV) richness was similar across the three zones, however, the bacterial community composition and soil physicochemistry differed across the intact and degraded zones. We identified, via correlation, the potential drivers of microbial community composition as soil density, pH and the ratio of Ca to Mg. The restored zone was intermediate between the intact and degraded zones. The differences in the microbial communities appeared to be driven by the presence of plants, with plant-associated taxa more common in the intact zone. The dominant taxa in the degraded zone were cosmopolitan organisms, that have been reported globally in a wide variety of habitats. This study provides baseline information on the changes of the soil bacterial community of a spatially restricted and threatened biome. It also provides a starting point for further studies on community composition and function concerning the restoration of degraded succulent thicket ecosystems.

## INTRODUCTION

The Albany Subtropical Thicket is a biome unique to South Africa that possesses a rich floristic diversity (*Hoare et al., 2006*); it consists of various forms of closed canopy shrubland, less than three meters in average canopy height, that grade into forest above ~800 mm annual precipitation and into karroid shrubland below ~200 mm

(*Vlok, Euston-Brown & Cowling, 2003*). Here we focus on an arid thicket type (100–500 mm mean annual precipitation) where large succulent shrubs, particularly *Portulacaria afra* (commonly known as "spekboom"), dominate the canopy (*Vlok, Euston-Brown & Cowling, 2003*). The succulent-rich thicket types have been noted for their impressive carbon storage, given the semi-arid climates in which they occur (*Mills & Cowling, 2010*; *van der Vyver & Cowling, 2019*). This biome predominantly occurs in the Eastern Cape region and extends into the Western Cape province, constituting 2.5% of the land area of South Africa (*Cowling et al., 2005*)—the "arid" and "valley" thicket structural types, where *P. afra* can be abundant, comprise over 50% of the biome (*Vlok, Euston-Brown & Cowling, 2003*; *Dayaram et al., 2019*). The Albany Subtropical Thicket is restricted to deep, well-drained, fertile, sandy loams with the densest thickets occurring in the deepest soils (*Cowling, 1983*; *Vlok, Euston-Brown & Cowling, 2003*) and occurs in semi-arid regions where there is sufficient protection from frost and fire (*Duker et al., 2015a*, *2015b*; *Cowling & Potts, 2015*).

*Portulacaria afra* is considered to be an ecosystem engineer in the arid and valley thicket subtypes where it is dominant (*Lechmere-Oertel, Cowling & Kerley, 2005*; *Lechmere-Oertel et al., 2008*; *van Luijk et al., 2013*). These regions are referred to as "succulent thicket" (*sensu Moolman & Cowling (1994)*). This stem-succulent shrub produces an unusually large biomass for the arid environment in which it grows. This has been attributed to its ability to shift between the C3 and crassulacean acid metabolism photosynthetic pathways (*Ting & Hanscom, 1977*; *Guralnick & Ting, 1987*), that likely enables it to take advantage of sporadic rainfall (*Mills et al., 2014*), store large quantities of carbon and maintain metabolic activity during drought conditions by recycling organic acids (*Guralnick, Rorabaugh & Hanscom, 1984*). Consequently, *P. afra* produces copious leaf litter and root biomass, generating soils with a high soil carbon content (*Mills et al., 2005*; *Lechmere-Oertel et al., 2008*; *Mills & Cowling, 2010*; *van der Vyver & Cowling, 2019*) that enhances local soil fertility (*Mills et al., 2005*) and soil moisture retention (*van Luijk et al., 2013*). In addition, the thick litter layer produced by *P. afra* improves soil moisture retention (*van Luijk et al., 2013*) by buffering wet and dry cycles, thus creating a favourable environment for other plant species (*Sigwela et al., 2009*; *van Luijk et al., 2013*; *Wilman et al., 2014*).

Extensive *P. afra* removal results in a shift to an alternative stable state, where the resulting ecosystem can be similar to that found in other regions (*van Luijk et al., 2013*) such as the Nama Karoo, open savanna or pseudo-savanna (*Lechmere-Oertel, Kerley & Cowling, 2005*; *Mills et al., 2005*). In degraded areas, soil organic carbon content is substantially reduced, as is water infiltration, resulting in lower water retention and increased erosion (*van Luijk et al., 2013*). The reduction in soil carbon content in degraded thicket habitat can be attributed to diminished carbon input from leaf litter and roots (*Mills & Cowling, 2006*), capping, and loss of topsoil through erosion (*Mills & Fey, 2004a*). It has also been suggested that processes such as increased microbial activity from elevated soil temperatures (*Jenkinson, 1981*) and increased wetting and drying cycles in exposed surface soil will increase the rate of soil organic matter mineralisation (*Birch, 1958*). Degraded succulent thicket does not spontaneously regenerate even in the absence

of herbivory (*Lechmere-Oertel, Kerley & Cowling, 2005*; *Lechmere-Oertel, Cowling & Kerley, 2005*).

Approximately 60% of the Albany Subtropical Thicket biome has been severely degraded (*Lloyd, Berg & Palmer, 2002*) by vegetation clearing, cutting of wood and, primarily, browsing by domestic herbivores. Only 11% of the thicket's original range remains intact (*Lloyd, Berg & Palmer, 2002*) with the rest either transformed or moderately degraded. Attempts have been made to restore the biodiversity and functionality of this ecosystem by replanting *P. afra* cuttings. These attempts have met with varying degrees of success; some areas have become revegetated (*Mills & Cowling, 2006*; *van der Vyver, Mills & Cowling, 2021*), while others have high mortality or low growth rates (average 28% survival) (*Mills & Robson, 2017*). It has been suggested that soil microbial diversity plays an important role in maintaining soil microbiome stability during periods of stress and recovery (*Garbeva et al., 2006*) and this may be the case in intact Albany Subtropical Thicket.

The relationship between terrestrial macroorganisms and microorganisms in the soil is an important component in understanding the structure and function of any ecosystem. Microorganisms perform important ecosystem services (*Bardgett & van der Putten, 2014*), including organic matter decomposition, nutrient recycling, fertility promotion and soil agglomeration (*Xun et al., 2018*). Factors influencing soil bacterial communities include physicochemical properties, organic matter content, fertilizer treatment, land-use, water availability and climate change (*Brodie, Edwards & Clipson, 2002*; *Marschner, 2003*; *Grayston et al., 2004*; *Ulrich & Becker, 2006*; *McCrackin et al., 2008*; *Jansson & Hofmockel, 2020*).

Despite the potential importance of microbial communities in the establishment and maintenance of the Albany Subtropical Thicket, neither the edaphic microbiomes of this region nor the impact of *P. afra* removal on soil microbiome functioning has been characterised. Thus, the aim of this study is to compare the compositions of the soil microbiomes between intact (vegetated), degraded and partially-restored succulent thicket zones and identify environmental factors that could account for observed changes in the microbial community resulting from the loss and restoration of the succulent thicket vegetation.

## METHODS

### Sample acquisition and soil analysis

The study site, of approximately 55,000 m$^2$ (Fig. 1), was located in the Eastern Cape, South Africa (33.2977° S, 24.7461° E). Sampling was performed on 12 December 2019 along six parallel transects with approximately 50 m between sample collection sites. The study site is bisected by a fence where half of the land area is in a degraded state (due to over-browsing by domestic animals over many decades) and half is intact (largely protected from excessive browsing). Fifteen soil samples (0–5 cm depth, after removal of surface leaf litter) were collected from each of the degraded and protected areas. In addition, five samples were taken from under the canopy of *P. afra* plants that were planted in the degraded area (February 2009), where a 50 m by 50 m area was fenced and a
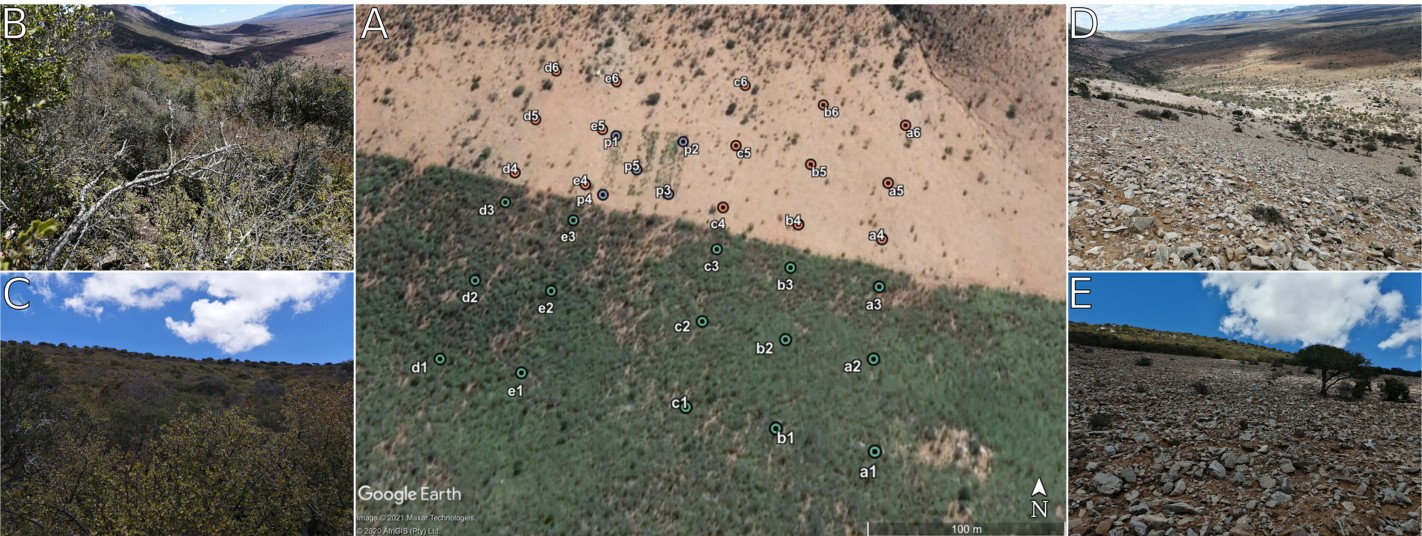

**Figure 1** **The layout and appearance of the study site.** (A) The layout of the study site as seen in Google Earth Pro. Sampling sites numbered 1–3 occur in the intact succulent thicket while sites 4–6 occur in the degraded succulent thicket. P1–P5 are in the restored zone. The side panels show photographs taken at (B + C) intact and (D + E) degraded sampling sites. See Fig. S1 for further historical imagery of the site.

range of *P. afra* planting treatments were trialled (as part of the large-scale restoration experiment detailed in *Mills et al. (2015)*). Here we term these various states of thicket vegetation as the following: the intact zone, the degraded zone, and the restored zone (although the defined area was only partially restored) and a "site" is the point where soil was sampled for microbial DNA extraction and soil physicochemical analysis. Soil samples were stored on ice immediately after collection and transferred to a −40 °C freezer within six h of sampling. Frozen samples were transferred from Port Elizabeth to Pretoria on ice and transfer took less than 8 h. In Pretoria, the samples were stored at −80 °C until DNA extraction.

For each of the vegetation conditions, three iButton data loggers (DS1923-F5# Hygrochron; iButtonLink, LLC, Whitewater, WI, USA) were placed at five cm soil depth and recorded the temperature and humidity every two h from 13 July 2020 to 6 December 2020.

Approximately 250 g of soil from each sample site was submitted to Intertek Agricultural Services for soil chemistry analysis (Intertek, Johannesburg, South Africa). The following properties were measured: soil pH (KCl), P (Bray 1/Bray 2), cations (Ca, Mg, K, Na, S) (Mehlich 3), exchangeable acidity, density, %Ca, %Mg, %K, %Na, Ca:Mg, Ca + Mg/K, texture (Clay, Silt, Sand), total organic carbon (Walkley Black), $NH_4$-N and $NO_3$-N.

## DNA Extraction and Sequencing

Metagenomic DNA (*i.e.*, DNA found in an environmental sample) was extracted from 0.5 g soil samples using the QIAGEN DNeasy PowerSoil kit (Qiagen, Venlo, Netherlands). The protocol was modified to include an additional step of soil agitation by two 40 s
cycles of 2,500 rpm in a Powerlyzer 24 (Qiagen, Venlo, Netherlands). We evaluated the quantity and quality of the DNA with a Nanodrop 2000 spectrophotometer (Thermo Fisher Scientific, Waltham, MA, USA) and by amplifying the bacterial 16S rRNA V3–V4 region (E9F and U1510R primers) using OneTaq® Hot Start DNA Polymerase (New England Biolabs, Ipswich, MA, USA). Metagenomic DNA was submitted to Omega Bioservices (Norcross, Georgia, United States) for sequencing of the V3–V4 region of the 16S rRNA gene (Forward primer: 5′-CCTACGGGNGGCWGCAG-3′, reverse primer: 5′-GACTACHVGGGTATCTAATCC-3′ (*Klindworth et al., 2013*)) on an Illumina MiSeq v3 with paired-end 300 bp reads. Each sample was sequenced twice.

## Data analysis

The raw DNA reads were processed in QIIME2 2020.8 (*Bolyen et al., 2019*), trimming 15 bp off the start and end of the reads and denoised using DADA2 (*Callahan et al., 2016*) to give amplicon sequence variants (ASVs) (*Callahan, McMurdie & Holmes, 2017*) which identify different bacterial sequences with single nucleotide accuracy. After processing, library sizes ranged from 36,067 reads to 116,915 reads with a mean of 87,305 reads and a median of 90,359 reads. Each ASV was assigned taxonomy by comparing the sequence to the SILVA 138.1 (*Quast et al., 2013*; *Yilmaz et al., 2014*) database of 16S rRNA gene sequences using a naive Bayes classifier.

Unless otherwise stated, all data were analysed in R 4.0.3 (*R Core Team, 2020*) using the following packages and their dependencies: Phyloseq 1.34.0 (*McMurdie & Holmes, 2013*), ggplot2 3.3.2 (*Wickham, 2016*), stringr 1.4.0 (*Wickham, 2019*), pheatmap 1.0.12 (*Kolde, 2019*), RcolorBrewer 1.1-2 (*Neuwirth, 2014*), vegan 2.5-6 (*Oksanen et al., 2019*), gridExtra 2.3 (*Auguie, 2017*), NetCoMi 1.0.2.9000 (*Peschel et al., 2020*), lubridate 1.7.9.2 (*Grolemund & Wickham, 2011*), ggrepel 0.9.1 (*Slowikowski, 2021*) and ggsignif 0.6.0 (*Ahlmann-Eltze, 2019*). Except for alpha diversity, which was calculated on unnormalised data, all analyses used ASV data transformed for relative abundance (proportions) which have the best performance for community analysis (*McKnight et al., 2019*). Beta diversity was calculated with the quantitative Jaccard metric. The Principal Co-ordinates Analysis (PCoA) used all the available ASVs but for other analyses, ASVs were agglomerated at either the phylum or genus level, as specified in the results where applicable. For the Principal Components Analysis (PCA) and Redundancy Analysis (RDA), given that data were obtained from three iButtons per zone, we interpolated the available data to generate pseudomeasurements for each site. The interpolation was performed by randomly drawing each variable from a normal distribution with a mean and standard deviation appropriate for each zone. In addition, the soil physicochemical values were standardised to zero mean and unit variance. The appropriate model for the RDA was chosen by including the terms which were selected by automatic stepwise model building using the functions ordistep and ordiR2step from the R package vegan. The final model was tested and evaluated by Anova to ensure that all terms were statistically significant. The co-occurrence network was constructed with associations calculated with CCREPE (also known as ReBoot) (*Faust et al., 2012*) and called from NetCoMi with the default parameters and clustered with the default "cluster_fast_greedy"

algorithm (*Clauset, Newman & Moore, 2004*). Linear discriminant analysis Effect Size (LEfSe) (*Segata et al., 2011*) was used to identify the bacterial taxa that were differentially abundant between sites. We used the LEfSe implementation on the Huttenhower Lab Galaxy Server: https://huttenhower.sph.harvard.edu/galaxy/. LEfSe uses the relative abundance of the ASVs, normalised so that the ASVs counts sum to one million in each sample, applies a Kruskal-Wallis test to identify features with a significant difference between the sample sets and uses Linear Discriminant Analysis to estimate effect sizes; finally returning biomarkers where the effect size has a logarithmic score (base ten) greater than two and the *p*-value is less than 0.05.

The sample metadata are provided in the Supplemental Files (Table S1) and all scripts used for analysis are available on GitHub: https://github.com/jasonbosch/The-soil-microbiomics-of-intact-degraded-and-partially-restored-semi-arid-succulent-thicket.

# RESULTS AND DISCUSSION

## Soil physicochemistry

This study is based on a detailed comparison of prokaryotic microbial diversity in the 0–5 cm soil horizon from two closely adjacent but substantially different habitats: an intact (a largely undisturbed and vegetated) succulent thicket zone, and a degraded zone, where decades of unsustainable browsing resulted in the near complete removal of vegetation (Fig. S1) with subsequent erosion of topsoil (with bedrock evident in places). Samples taken from a partially revegetated area (the restored zone) were also included to assess the impact of (partial) restoration on the soil after approximately a decade of *P. afra* planting.

The two comparative areas, the intact and degraded zones, might be expected to differ in both biotic and abiotic parameters due to the widely different vegetation cover of the two areas (mature succulent thicket *vs* a sparse herbaceous layer) (Fig. 1). Specifically, the increased litter inputs and shading in the undisturbed vegetated area is predicted to increase carbon input and moisture retention in the soil (*van Luijk et al., 2013*), thereby positively impacting the soil microbial communities. By comparison, degraded areas are exposed to direct sunlight and wind; both of which are predicted to decrease water retention by increasing evapotranspiration and to negatively impact microbial communities. However, the degraded areas may also positively benefit, in highly localised patches, from the presence of domestic animals, specifically from the input of urine and faeces, which might also have an impact on the microbial communities in these zones (*Todkill, Kerley & Campbell, 2006*).

Physicochemical analyses of the three primary experimental zones (intact, degraded and restored) (Table S1) showed significant differences in several parameters and intact and degraded zone samples clustered separately when analysed *via* PCA (Fig. 2A). The intact succulent thicket soil samples had a significantly higher total organic carbon content than either the restored or degraded zone soils (Intact: 2.33 ± 1.00, Degraded: 0.72 ± 0.28, Restored: 1.18 ± 0.15; %; Wilcox test: Intact v Degraded: $p = 7.71 \times 10^{-5}$, Intact v. Restored: $p = 2.27 \times 10^{-2}$, Degraded v. Restored: $p = 4.30 \times 10^{-3}$). This can be attributed to the input of leaf litter from the vegetated cover. *P. afra*, the main component of

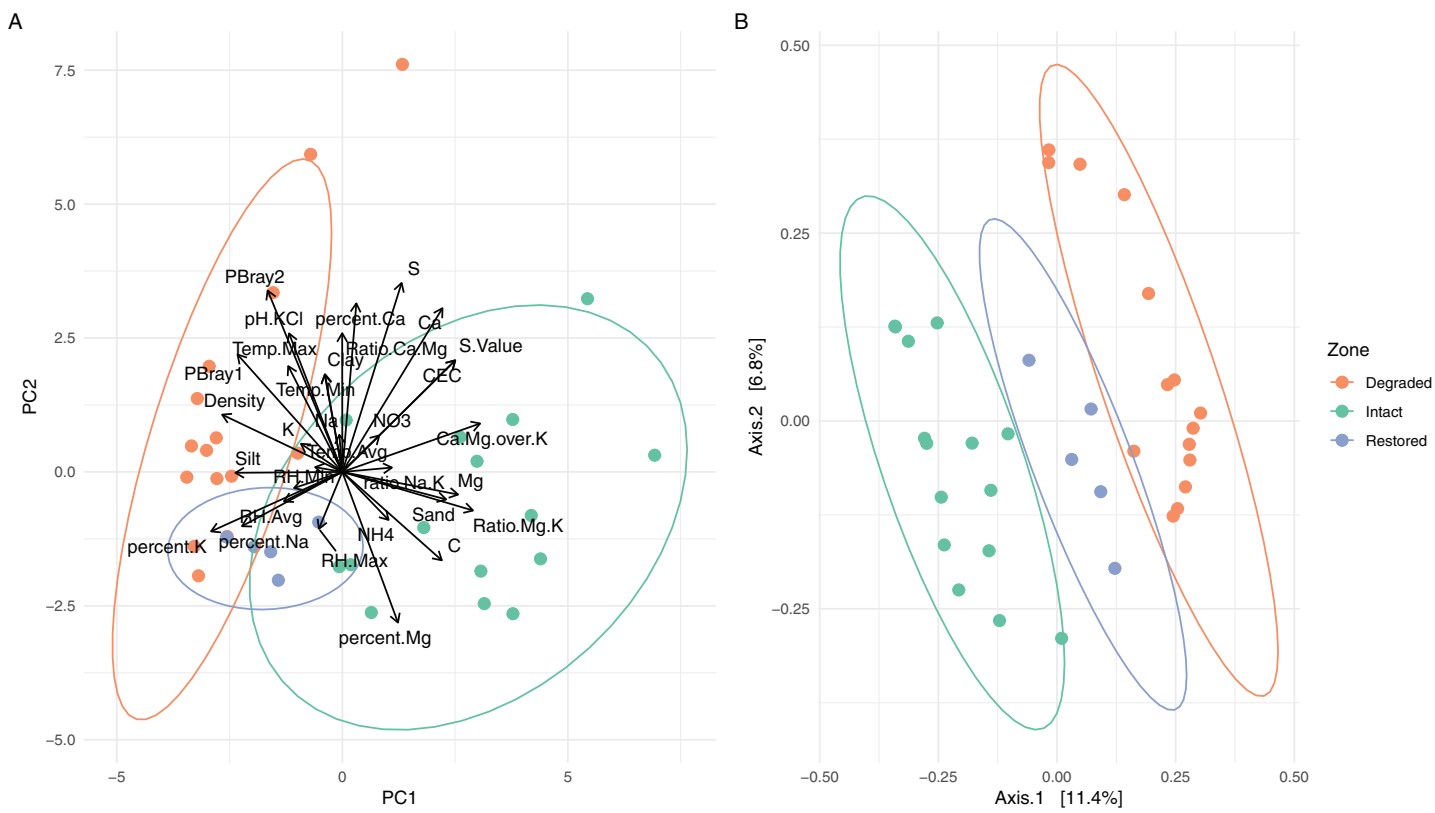

**Figure 2 Comparison of the different vegetation zones through Principal Components Analysis (PCA) and Principal Co-ordinates Analysis (PCoA).** (A) Principal Components Analysis (PCA) of the soil physicochemistry demonstrating a separation between the intact and degraded soils with restored soils overlapping with the other two zones. The variables, their relative weights and directions are shown as black arrows. The first two principal components contain 45.36% of the variation. (B) Principal Co-ordinates Analysis (PCoA) using weighted Jaccard distances based on ASV composition showing that the samples from different zones group together without overlap. When grouped by zone, the PCoA explains 15.13% of the variation.

succulent thicket vegetation, is known to create a carbon-rich soil environment (*Lechmere-Oertel et al., 2008*; *Mills & Cowling, 2010*; *van der Vyver & Cowling, 2019*). The intact zone samples had the highest measured levels of $Ca^+$ (Intact: 3,174.20 ± 1,374.18, Degraded: 2,168.87 ± 1,789.83, Restored: 1,773.20 ± 396.15; mg/kg) and the lowest pH (Intact: 5.48 ± 0.90, Degraded: 6.84 ± 0.75, Restored: 6.53 ± 0.33), phosphorous content (PBray1: Intact: 18.53 ± 6.06, Degraded: 44.07 ± 9.14, Restored: 31.60 ± 9.81; PBray2: Intact: 27.33 ± 11.29, Degraded: 71.4 ± 28.15, Restored: 52.60 ± 9.74; mg/kg) and bulk density (Intact: 0.94 ± 0.12, Degraded: 1.32 ± 0.08, Restored: 1.15 ± 0.07; g/ml). In comparison, the degraded zone soils showed the lowest $Mg^{+2}$ levels (Intact: 1,038.73 ± 580.19, Degraded: 368.93 ± 180.56, Restored: 789.40 ± 187.38; mg/kg) and cation exchange capacity values (Intact: 26.42 ± 10.32, Degraded: 16.03 ± 10.95, Restored: 18.70 ± 4.12), possibly a reflection of the higher water infiltration rates in the largely unvegetated exposed soils. Unexpectedly, we did not observe significant differences in $NH_4–N$ (Intact: 51.75 ± 31.63, Degraded: 38.32 ± 7.34, Restored: 41.66 ± 6.51; mg/kg; Wilcox test: Intact v. Degraded: $p = 0.74$, Intact v. Restored: $p = 0.55$, Degraded v. Restored: $p = 0.17$) or $NO_3–N$ content (Intact: 32.18 ± 37.97, Degraded: 24.03 ± 16.89, Restored: 13.30 ±

7.96; mg/kg; Wilcox test: Intact v. Degraded: $p = 0.59$, Intact v. Restored: $p = 0.30$, Degraded v. Restored: $p = 0.23$) between samples from the different zones. These results replicate previous findings for pH and C (*Mills & Fey, 2004b*; *Lechmere-Oertel, Cowling & Kerley, 2005*) as well as for *P*, silt and Mg but not for Ca and $NH_4$-N (*Mills & Fey, 2004b*).

As expected, the loss of succulent thicket had a major impact on soil temperature buffering and moisture entrapment, which was reflected in significant differences of temperature (Intact: $19.73 \pm 7.12$, Degraded: $21.41 \pm 9.45$, Restored: $19.57 \pm 4.95$; °C; Wilcox test: Intact v. Degraded: $p = 1.07 \times 10^{-9}$, Intact v. Restored: $p = 3.94 \times 10^{-9}$, Degraded v. Restored: $p = 0.02$) and relative humidity (Intact: $53.18 \pm 21.40$, Degraded: $58.13 \pm 24.83$, Restored: $60.04 \pm 20.73$; %; Wilcox test: Intact v. Degraded: $p = 2.11 \times 10^{-10}$, Intact v. Restored: $p = 9.32 \times 10^{-63}$, Degraded v. Restored: $p = 8.47 \times 10^{-24}$) between the three experimental zones (Table S2 and Figs. S2–S4). The degraded zone had significantly higher maximum daily temperatures than either the intact or restored zones (Intact: $34.40 \pm 8.18$, Degraded: $36.66 \pm 8.68$, Restored: $30.03 \pm 7.43$; °C; Wilcox test: Intact v. Degraded: $p = 0.03$, Intact v. Restored: $p = 6.55 \times 10^{-7}$, Degraded v. Restored: $p = 3.21 \times 10^{-11}$; Fig. S3), most likely due to the soil being directly exposed to sunlight, in agreement with observations from similarly degraded thicket landscapes (*Lechmere-Oertel et al., 2008*). The lower maximum soil temperatures in the restored zone compared to the intact zone (Fig. S3) may be due to greater localised canopy closure which would buffer soil temperatures.

The degraded zone originally displayed a lower daily maximum relative humidity when compared to the intact zone. Following the onset of seasonal rains, the degraded zone exhibited higher daily maximum and minimum relative humidity (Fig. S4). This is unsurprising as there is high rainfall interception in intact thicket canopy (~60% interception, amongst the highest values recorded for various vegetation types across the globe (*Cowling & Mills, 2011*; *van Luijk et al., 2013*)) and thus smaller rainfall events have very little impact on soil moisture beneath the intact thicket canopy relative to the bare ground in the degraded zone. Secondly, the deep lens of low bulk density soil under intact thicket means that water rapidly infiltrates beyond five cm (*van Luijk et al., 2013*). Thus, the soil moisture under the intact canopy was lower than in the degraded zone for small rainfall events, but water will likely be stored in the leaf litter lens after large rainfall events extending the period of water availability (*van Luijk et al., 2013*). We suspect that only small (<5 mm) rainfall events occurred during the period the iButtons were deployed. The restored site lacked the deep litter lens, exhibited a layer of silt trapped from the degraded area and had a more closed canopy than the intact zone, together ensuring that soil relative humidity values were higher than in the degraded and intact zones (Fig. S4).

## Biodiversity and microbial composition differences between the intact and degraded zones

Alpha-diversity analysis of the 26,759 observed ASVs revealed no significant differences in biodiversity between intact and degraded zones (Fig. S5). This was unexpected and

contradicts a recent study that found higher levels of diversity in arid soils with plant cover compared to those without (*Kushwaha et al., 2021*). However, previous studies have disagreed whether herbivory increases (*Eldridge et al., 2017*) or decreases (*Cheng et al., 2016*) bacterial diversity and the link between browsing and soil microbial diversity is probably very complex and may depend on both browsing intensity and the plant community species diversity and composition (*Qu et al., 2016*). Despite the similarities in biodiversity summary statistics, microbial communities from the different zones formed distinct clusters, as indicated by the beta-diversity distances between the samples (Fig. 2B and Fig. S6). Clustering of the microbial populations for the three zones (intact, degraded and restored) captured 15.13% of the variation in the samples, with restored samples located in an intermediate position between the intact and degraded samples. Together these results suggest that vegetation loss has an impact on the community structure of the succulent thicket soil microbiome, but not on its overall biodiversity.

The core microbial community (defined as ASVs present in at least 95% (*i.e.*, ≥33/35) of samples: Table S3) only accounted for 103 genus-level ASVs (9.87%) but comprised 70.29% of the sequence reads. If the threshold were raised to 100% (*i.e.*, 35/35 sites), then 60 genus-level ASVs, comprising 5.75% of the total genus-level ASVs and 54.81% of the reads, would be detected but if it were lowered to 89% (*i.e.*, 31/35 sites) then 142 genus-level ASVs comprising 13.60% of the total genus-level ASVs and 77.84% of the reads, would be detected. Each of the three sampled zones also had their own unique core communities which showed a level of similar dominance across the reads (Tables S4–S6); the core community accounted for 118 ASVs (13.58% of genus-level ASVs, 74.95% of reads) in the intact zone, 166 ASVs (22.10% of genus-level ASVs, 88.43% of reads) in the degraded zone and 207 ASVs (30.49% of genus-level ASVs, 86.40% of reads) in the restored zone. If the threshold were raised to 100% (*i.e.*, 15/15 or 5/5 sites) then the core community would consist of 78 ASVs (8.98% of genus-level ASVs and 62.75% of reads) in the intact zone and 121 ASVs (16.11% of genus-level ASVs and 80.85% of reads) in the degraded zone while the restored zone would be the same as at the standard 95% threshold. If the threshold were lowered to 89% (*i.e.*, 13/15 or 4/5 sites) then the core community would consist of 155 ASVs (17.84% of genus-level ASVs and 81.68% of reads) in the intact zone, 189 ASVs (25.16% of genus-level ASVs and 90.77% of reads) in the degraded zone and 295 ASVs (43.45% of genus-level ASVs and 93.74% of reads) in the restored zone. The dominance of a relatively small number of taxa is a well-known phenomenon in soils (*Delgado-Baquerizo et al., 2018*). The composite microbial community in all experimental samples comprised 36 prokaryotic phyla, the most abundant of which (based on ASV assignments of 16S rRNA gene amplicon reads) were Actinobacteriota (28.76%), Proteobacteria (21.39%), Acidobacteriota (11.40%), Plantomycetes (9.84%) and Bacteroidota (7.71%) (Fig. 3A).

Significant differences were observed at the genus level between the intact and degraded zones (Fig. 3B and Fig. S7). The largest differences were observed for *Rubrobacter* (1.72% intact *vs* 6.16% degraded), *Conexibacter* (3.57% intact *vs* 0.14% degraded), RB41 (2.22% intact *vs* 5.02% degraded), *Bryobacter* (4.45% intact *vs* 1.82% degraded) and *Mycobacterium* (2.69% intact *vs* 0.53% degraded). ASVs with a relative abundance of

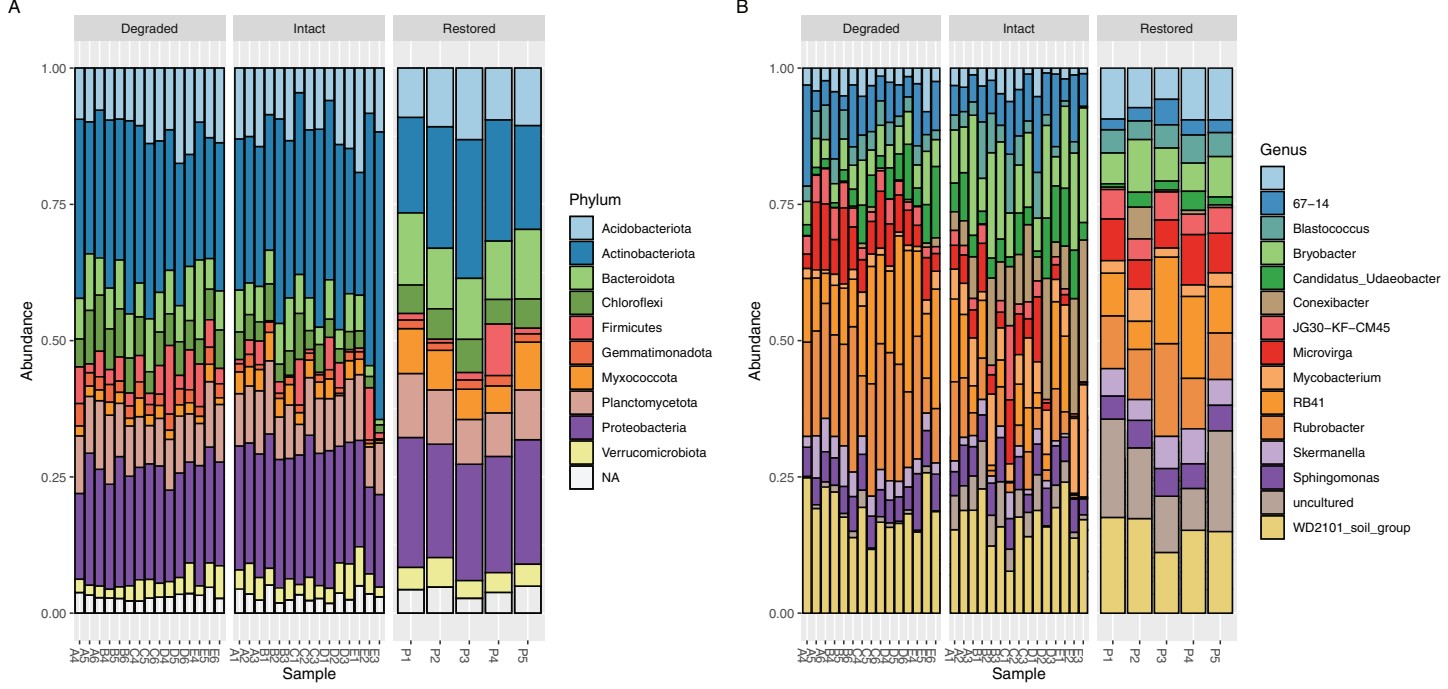

**Figure 3** **Relative abundance of bacterial taxa.** Relative abundance of (A) bacterial phyla and (B) genera. Taxa which have a relative abundance less than 1% are combined in the NA category in (A) and are completely removed in (B) for ease of viewing. The taxa displayed in (B) account for approximately 25% of the total abundance in the samples.

less than 1% accounted for between 66.05% and 69.94% of all reads in the various zones. Soil samples from the restored zone showed a larger number of taxa classified as 'uncultured' than either the intact or degraded zone samples (4.06% restored *vs* 1.50% intact and 0.29% degraded).

In addition, LEfSe analysis identified four biomarker taxa for intact zone soils and five for degraded zone soils (Fig. 4). The intact biomarker taxa were derived from the families *Acidobacteriaceae (Subgroup 1)* and *Myxococcaceae*, the order Frankiales and the class Verrucomicrobiae. The order Frankiales is one of the most abundant in the dataset and includes many root-nodule associated taxa (*Pawlowski & Demchenko, 2012*; *Battenberg et al., 2017*); its over-representation in intact sites suggests that the changes in the soil microbiome may be due to the reduction or disappearance of plant-associated taxa with the loss of vegetation. By comparison, biomarker taxa for degraded zone soils were the genera *Ensifer* and E*xiguobacterium*, members of which are found in diverse environments (*Kasana & Pandey, 2018*), the cyanobacterial family *Coleofasiculaceae*, the order Puniceispirales and the Chloroflexi class Anaerolineae, commonly found in anaerobic digesters (*Xia et al., 2016*).

## Abiotic drivers of microbial community structure in both intact and degraded zones

In order to determine which soil physicochemical properties were potentially important for microbial community structure, we used RDA (Fig. 5). At the phylum level, the ratio of

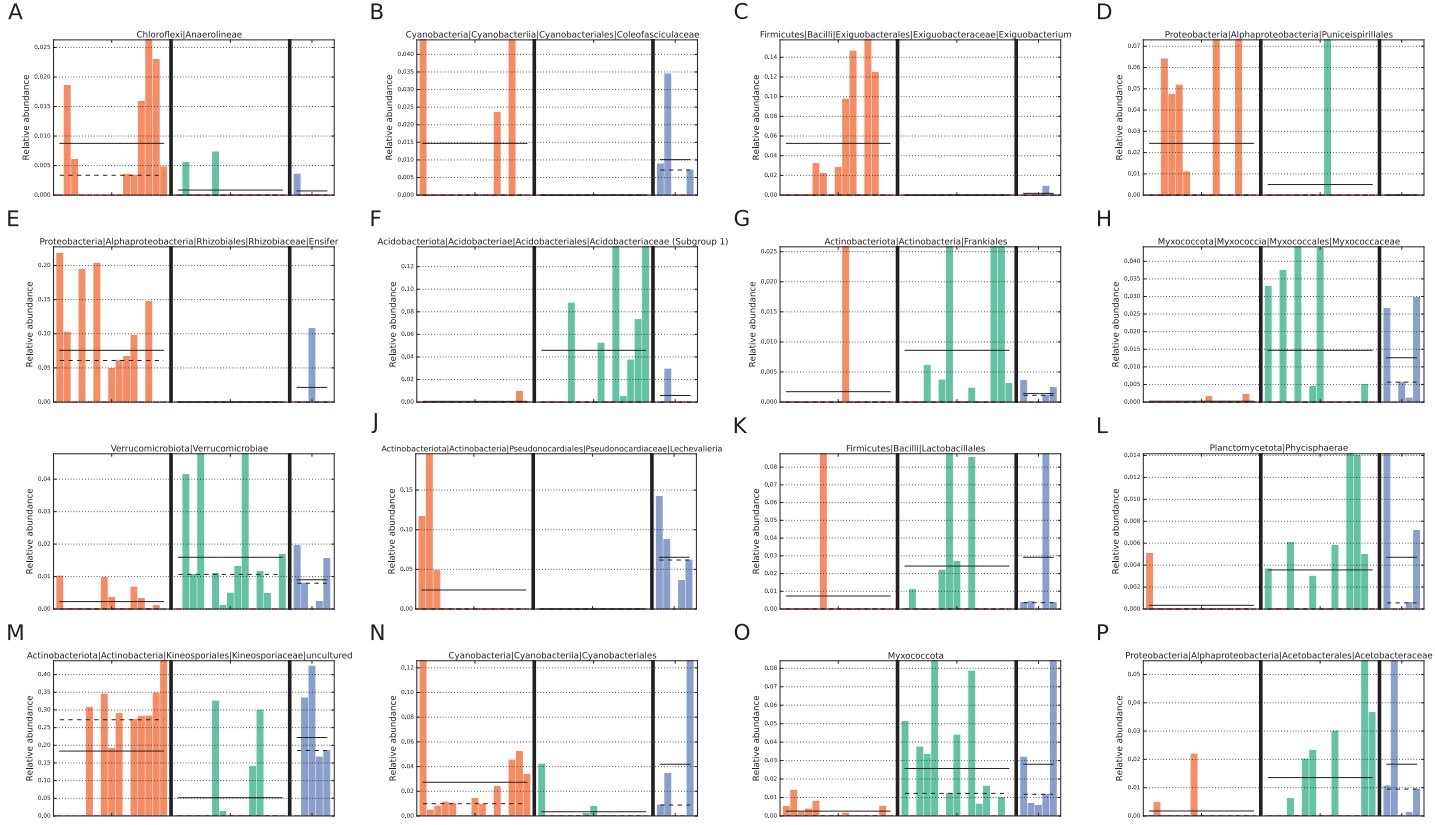

**Figure 4 Taxa identified as potential biomarkers.** Taxa identified as potential biomarkers according to LEfSe analysis. Only the lowest taxa in the hierarchy is displayed for a particular condition. Taxa were identified as biomarkers for (A–E) degraded vegetation, (F–I) intact vegetation or (J–P) restored vegetation. Colours represent the different zones; orange = degraded, green = intact, blue = restored.

Ca to Mg and the soil density explained 47.78% of the variation in the microbial community (Fig. 5A) and, at the genus level, the ratio of Ca to Mg, the soil density and the soil pH were able to explain 50.10% of the variation in the community structure (Fig. 5B, a third axis is not plotted). pH has frequently been identified as a major driver of bacterial community composition in soils (*Rousk et al., 2010*; *Qu et al., 2016*). Contrary to one of the initial expectations of this study, differences in soil relative humidity and temperature resulting from the loss of vegetation did not appear to significant affect microbial community structure in the intact and degraded zones. However, the interplay of relative humidity and temperature may affect the water balance of the soil, which could potentially be responsible for the shift in pH (*Slessarev et al., 2016*).

## Unique to near-unique members of the core microbial community in each zone

To identify unique members of the common core microbial community (*Risely, 2020*) within each zone, genus-level ASVs were filtered using the following two criteria: the ASVs were present in >95% of the sites within a zone and in fewer than 10% of sites within the other zones (Table S7). The number of near-unique taxa in each zone were also tested

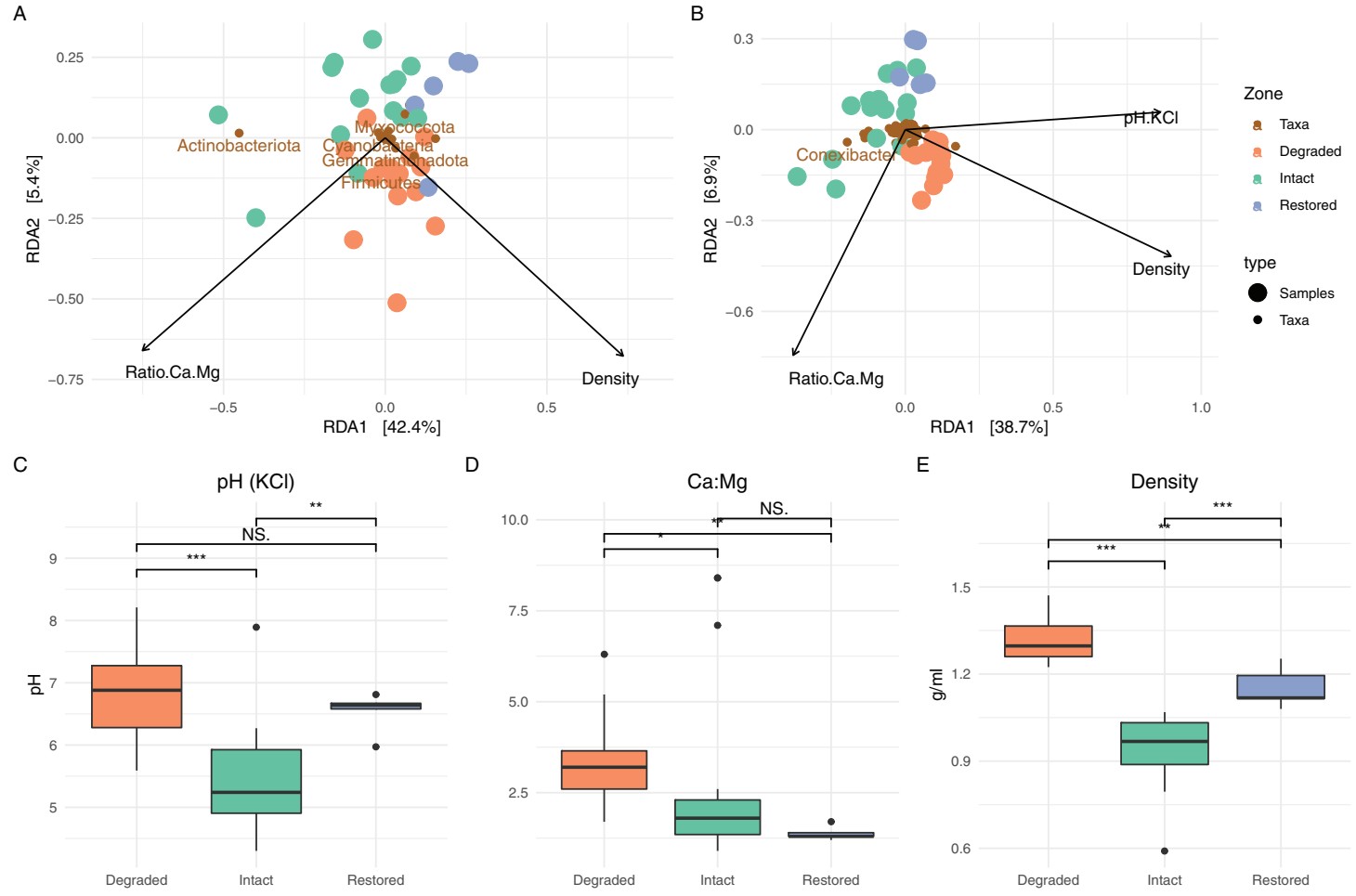

**Figure 5 Redundancy analysis and associated soil physicochemistry.** On the top row are the RDA results when assessing the communities at the (A) phylum and (B) genus level. Only a limited number of taxa names are displayed to prevent label overlap. The bottom row shows box-and-whisker plots of the variables which were determined to play a significant role in the RDA; (C) pH, (D) ratio of Ca to Mg and (E) density. Significance was determined using the Wilcox test.

at different thresholds to assess how the threshold affected the results (Table S8). The threshold for the number of sites in which an ASV had to present for a specific zone had the greatest effect on the results, while the threshold for the number of non-zone sites in which an ASV could be present only had an effect when the threshold was raised to 20% or 3/15 sites. While there were several unique core members for the restored zone, there were fewer sites for that zone and all the unique core members were at or below the mean relative abundance of detected taxa; for these reasons the unique core taxa of only the intact and degraded zone samples are discussed.

In the intact (vegetated) zone core community, two genera were unique: *Acidipila-silvibacterium* and *Burkholderia-caballeronia-paraburkholderia*. *Acidipila-silvibacterium* is a member of the Acidobacteriota, commonly found in soils and capable of tolerating tolerate low pHs (*Kielak et al., 2016*; *Kalam et al., 2020*). The family *Acidobacteriaceae (Subgroup 1)*, which contains *Acidipila-silvibacterium*, was also identified by LEfSe analysis as a biomarker of intact succulent thicket soil samples (Fig. 4), consistent with the

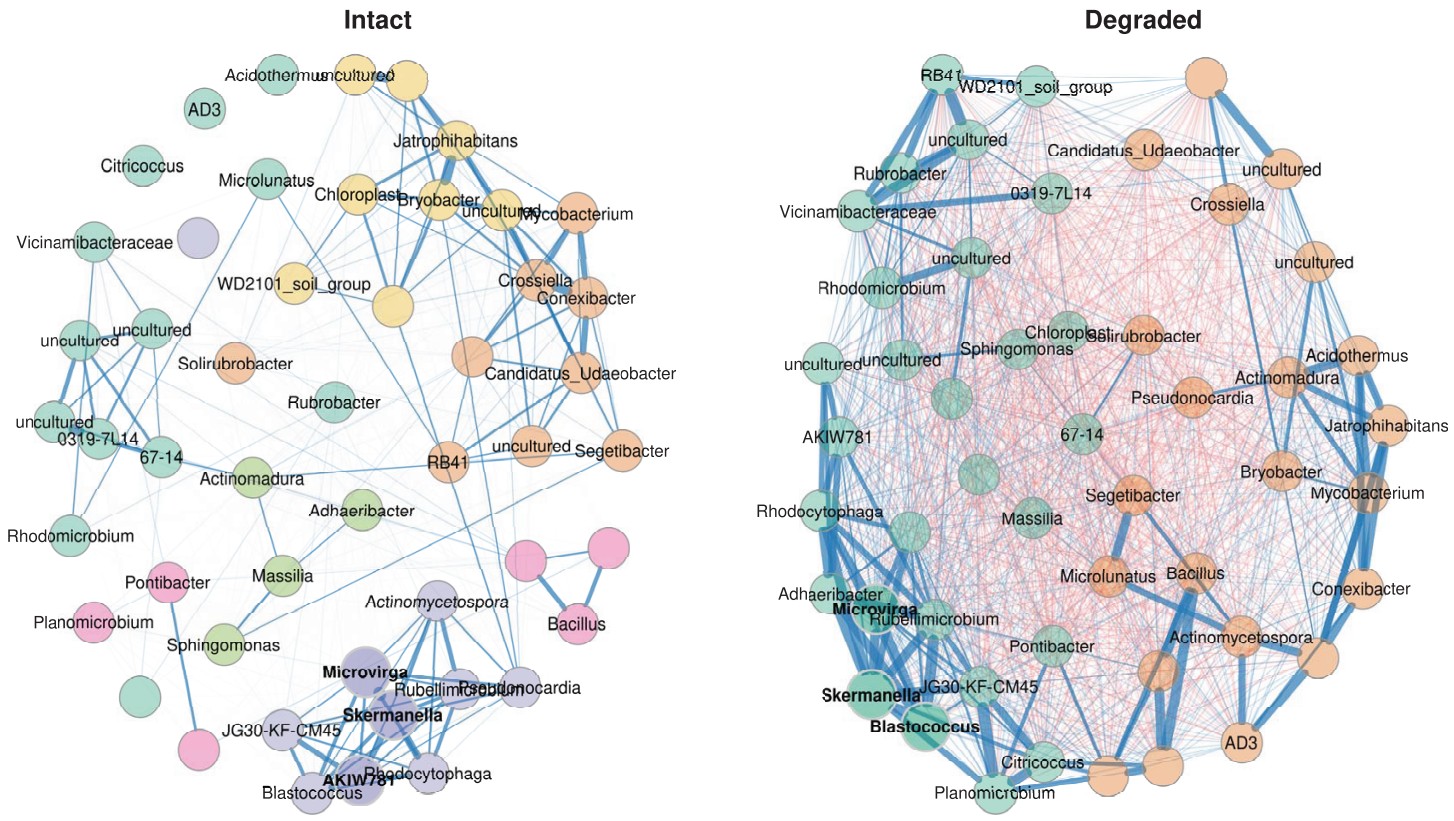

**Figure 6 Co-occurrence network.** Co-occurrence network showing the 50 ASVs with the highest variance. Bold text indicates hub nodes and nodes of the same colour were clustered together. Blue edges indicate positive correlations and red edges indicate negative correlations between the connected nodes. The thicker the edge, the more similar the two nodes are to one another.

lower mean pH values of these samples (Intact: pH 5.48, degraded: pH 6.84). It is likely that the presence of these taxa, almost exclusively in the vegetated soil samples, is due to the abundance of plant biomass, given that *Acidipila-silvibacterium* has been identified as a core operational taxonomic unit (OTU) of decaying wood (*Tláskal et al., 2017*) and *Burkholderia-caballeronia-paraburkholderia* contains many plant-associated species (*Compant et al., 2008*) which degrade cellulose (*Štursová et al., 2012*) and are associated with wood-decaying fungi (*Christofides et al., 2020*).

The two unique genera identified in the degraded zone core community were an uncultured member of the order Azospirillales and the genus *Arthrobacter*. Both of these taxa are potentially nitrogen-fixing (*Steenhoudt & Vanderleyden, 2000*; *Fernández-González et al., 2017*) and *Arthrobacter* has been implicated in the recovery of polluted soils and soils where vegetation has been lost (*Wang, Xie & Hu, 2013*; *Kim & Song, 2014*; *Fernández-González et al., 2017*).

### Correlations of taxa in the intact and degraded sites

In order to understand how the structure of the microbial communities changed between intact and degraded zones, co-occurrence networks of the 50 genus-level ASVs with the highest variation in abundance were filtered from the main dataset and compared

between the intact and degraded zones. The network analysis showed a marked decrease in community complexity in degraded compared with intact zone microbiomes (Fig. 6) as measured by clusters of co-occurring taxa. In the intact zone samples, ASVs were grouped in six clusters, whereas in the degraded zone samples, the same ASVs formed only two clusters. The cluster assignments were consistent, even when the number of genus-level ASVs were increased, indicating that the differences in complexity were robust for the ASVs included in the network.

In the intact zone network, only positive correlations are observed within groups; surprisingly negative correlations were completely absent. This suggests that the intact zone microbial community exists in a stable state, where each cluster of taxa may occupy a distinct niche and where inter-taxon competition is minimal. In stark contrast, the degraded zone network showed very large numbers of both positive and negative correlations between the two groups (Fig. 6). A potential ecological implication of this observation is that niches in the degraded zone are largely homogenised, resulting in high levels of inter-taxon competition.

A closer examination of the 50 genus-level ASVs showed that 30 had significantly different abundances between the intact and degraded zones (Fig. S8); 15 in *cluster 1* and 15 in *cluster 2*. Thirteen of the 15 nodes belonging to *cluster 1* had higher abundance in the intact zone and 13/15 nodes belonging to *cluster 2* had higher abundance in the degraded zone. The genus-level ASVs found to be higher in the intact zone generally belonged to taxa which have been reported to be plant-associated, such as *Connexibacter* (*Dong et al., 2018*; *Dobrovolskaya et al., 2020*), *Mycobacterium* (*Bouam et al., 2018*; *Pan et al., 2020*), *Pseudonocardia* (*Chen et al., 2009*; *Zhao et al., 2012*; *Li et al., 2012*) and *Microlunatus* (*Tuo et al., 2016*). We note that these plant-associated taxa were also present in the degraded zone where vegetation is largely absent. It is unclear whether their presence is due to associations with the sparse vegetation, the presence of species in the plant-associated taxon category which do not undergo obligate interactions with plants or bacteria from the intact zone being carried downhill into the degraded zone by rain water (*Abu-Ashour & Lee, 2000*; *Caillon & Schelker, 2020*).

Although there are known pitfalls in interpreting microbial co-occurrence networks (*Armitage & Jones, 2019*; *Carr et al., 2019*), we suggest that the presence of abundant vegetation (in the intact zone), and the existence of plant-and plant root-associated microbiome niches, likely underlie the observed differences in the two networks (Fig. 6). Plant-root associated niches such as the rhizoplane and rhizospheric zones provide spatial and physicochemical separation for their intrinsic microbial communities (*Ofek-Lalzar et al., 2014*; *Battenberg et al., 2017*; *Morella et al., 2020*); consistent with the well-discriminated clustering structure of the intact zone network, the limited number of inter-cluster correlations and the absence of negative correlations. Conversely, the loss of these defined niche structures in the largely unvegetated degraded zone appears to spatially homogenise the microbial community, leading to a weak clustering structure and a high level of inter-taxon competition.

### The intermediate position of the restored zone

The inclusion of the restored zone, where *P. afra* had been allowed to regrow, provided the opportunity to evaluate the recovery of the succulent thicket after an ~11 year interval. Soil samples from the restored zone showed a microbial community that was intermediate between the intact and degraded zones (Fig. 2B). The abundances of some taxa in the restored zone samples were also intermediate between those in the intact and degraded zone samples: *e.g., Rubrobacter* (Fig. S7). Similarly, a PCA of soil physicochemical properties showed that the restored zone overlapped with both the intact and degraded zones, while the latter two showed no overlap (Fig. 2A). In addition, soil physicochemical properties such as the amount of carbon and soil density (Fig. 5), both mediated by the presence of *P. afra*, were positioned at levels between those of the intact and degraded zones. However, for several other properties, the restored zone samples showed no statistically significant difference from those of the intact or degraded zone. This may indicate that different properties recover at different rates, but may also be due to stochastic variations between sites.

Taken together, these data suggest that the restored zone soils exist in an intermediate state between the intact and degraded zone soils. The obvious implication is that the planting of *P. afra* in degraded zones, as the basis of the restoration program, has resulted in a shift in both the soil properties and microbial communities, from the degraded state to more closely resemble the intact zone. To gain a full understanding of the process of restoration, multiple independent restoration attempts should be established in conjunction with regular, long-term monitoring in order to follow microbial succession (*Banning et al., 2011*) and distinguish between determined and stochastic events (*Zhou & Ning, 2017*). Understanding the temporal nature of community development, together with identification of the functionally important microbial species, would be an important aid to future restoration efforts (*Requena et al., 2001*; *Maestre, Solé & Singh, 2017*).

## ACKNOWLEDGEMENTS

The research presented here was part of a BSc (Hons) thesis (MS). We wish to thank Timm Hoffman for providing repeat photographs of the study site (used in Fig. S1).

### Funding

Jason Bosch and Jenny Johnson were supported by postdoctoral bursary funding from the University of Pretoria and Pedro Lebre was supported by senior postdoctoral bursary funding from the University of Pretoria. This work is based on research supported by the National Research Foundation of South Africa (Grant number: 119379; Alastair J. Potts) and the Department of Environment, Forestry and Fisheries: Natural Resource Management Programme (Robbert Duker; Alastair J. Potts). The funders had no role in study design, data collection and analysis, decision to publish, or preparation of the manuscript.

## Grant Disclosures

The following grant information was disclosed by the authors:
University of Pretoria.
National Research Foundation of South Africa: 119379.
Department of Environment, Forestry and Fisheries.

## Competing Interests

Alastair Potts is an Academic Editor for PeerJ and Don A. Cowan is a former member of the PeerJ Editorial Board.

## Author Contributions

- Micaela Schagen performed the experiments, analyzed the data, authored or reviewed drafts of the paper, and approved the final draft.
- Jason Bosch performed the experiments, analyzed the data, prepared figures and/or tables, authored or reviewed drafts of the paper, and approved the final draft.
- Jenny Johnson performed the experiments, authored or reviewed drafts of the paper, and approved the final draft.
- Robbert Duker performed the experiments, authored or reviewed drafts of the paper, and approved the final draft.
- Pedro Lebre performed the experiments, authored or reviewed drafts of the paper, and approved the final draft.
- Alastair J. Potts conceived and designed the experiments, prepared figures and/or tables, authored or reviewed drafts of the paper, and approved the final draft.
- Don A. Cowan conceived and designed the experiments, authored or reviewed drafts of the paper, and approved the final draft.

## Field Study Permissions

The following information was supplied relating to field study approvals (*i.e.*, approving body and any reference numbers):

No permit needed.

## DNA Deposition

The following information was supplied regarding the deposition of DNA sequences:

The raw sequencing data is available at the National Center for Biotechnology Information (NCBI) Sequence Read Archive (SRA): BioProject PRJNA735914.

## Data Availability

The iButton data is available at Zenodo: Cowan, Donald, Potts, Alastair, & Duker, Robbert. (2021). Thicket State Soil Temperature and Humidity Dataset: Thicket Soil Microbiome Project [Data set]. Zenodo. https://doi.org/10.5281/zenodo.4916017.

All scripts used for analysis are available at GitHub: https://github.com/jasonbosch/The-soil-microbiomics-of-intact-degraded-and-partially-restored-semi-arid-succulent-thicket.

## Supplemental Information

Supplemental information for this article can be found online at http://dx.doi.org/10.7717/peerj.12176#supplemental-information.

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
