# Peer review of "The soil microbiomics of intact, degraded and partially-restored semi-arid succulent thicket (Albany Subtropical Thicket)"

_PeerJ, doi:10.7717/peerj.12176_

## Round 0.1 · original submission · Major Revisions

We have received three reviews with important suggestions for improvement regarding methodology description and interpretation of results. I do not expect these comments to be difficult to address.

Reviewer 1 ·

Basic reporting

no comment

Experimental design

the fungal community should be also included in the study, especially it plays a very important role there.

Validity of the findings

no comment

Additional comments

The soil microbiomics of intact, degraded and partially-restored semi-arid succulent thicket (Albany Subtropical Thicket) by Schagen et al. reports the composition of soil microbiota between intact and degraded semi-arid succulent thicket. Albany Thicket biome is a typical ecosystem in South Africa and it is important to understand the soil microbiota that drive the nutrient cycling there.
Major comments:
1. Yes, soil microbiota includes bacterial and fungal communities. Why only bacterial community is examined in this study? If there is no specific reason for this, is it possible to add data of fungal communities?
2. As stated in lines 97-100 the aim of this study is to compare the composition of the soil microbiotas between the intact and degraded succulent thicket soils and identify environmental factors that could account for the observed changes in the microbial communities, the abstract and conclusion paragraph (lines 352-361) should include content of environmental factors.
3. lines 232-234 vegetation loss has no impact on the biodiversity. Could this be due to DNA extraction and PCR amplification biases? Was a “Mock community” included in the experiment?

Minor comments:
1. In the abstract, please state the aim of the study.
2. Please spread out the abbreviations “CAM” and “RDA” in their first appearance.
3. Data Analysis is poorly organized. Please describe each analysis step especially those in lines 143-149 in detail.
4. https://github.com/jasonbosch/The-soil-microbiomics-of-intact-degraded-and-partially-restored-semi-arid-succulent-thicket is not available. 16S rRNA data should be submitted to NCBI.
5. line 314 “A potential ecological implication of this observation are that niches”, “are” should be “is”?

·

Basic reporting

The manuscript entitled "The soil microbiomics of intact, degraded and partially-restored semi-arid succulent thicket (Albany Subtropical Thicket)" brings important considerations about the soil microbiota of impacted areas with pristine and reclaimed areas. I believe the manuscript raises interesting data. However I believe the authors can still value their results with some multivariate statistical analysis between abiotic factors (P and N) and microbial diversity (as suggested below)

Experimental design

1) At what temperature was the soil transported to the laboratory? And the storage conditions during the period prior to DNA extraction should also be described in Methods.
2) Describe the protocol used for chemical soil analysis. Another option would be to provide the reference that describes the chemical analysis procedures.
3) Provide the sequences of the primers used in the amplifications or provide the references
4) How many sequencing were performed for each site? Please make this information clear here in Methodos, you in Results address data such as reads and others, but the sequencing number of each sample must appear here for other scientists to follow/compare your methodology.
5) Describe which software and methods were used for the multivariate analysis (PCA and RDA).

Validity of the findings

1) In the section “Abiotic drivers of microbial community structure in both intact and degraded zones.” I believe that the authors should pay special attention to the levels of P and N as factors that influence diversity, these factors had influences in other studies. I strongly suggest that you read and cite the study: de Souza, L.C., Procópio, L. The profile of the soil microbiota in the Cerrado is influenced by land use. Appl Microbiol Biotechnol 105, 4791–4803 (2021).doi.org/10.1007/s00253-021-11377-w

2) Dear reviewers, I confess that I was unable to open your figures that would describe alpha-diversity. But anyway I believe that a picture describing comparing the abundance and diversity data could help in understanding the little difference between the two sites. It is common that when there is a loss of diversity, there is an increase in abundance. I believe their alpha diversity data could be further leveraged and discussed, including in the section "Unique to near-unique members of the core microbial community in each zone." And raise questions such as: these genera were more abundant in which place? The species of these genera were more diversified in which site?

Additional comments

The text is well written and the results are interesting and worthy of being discussed by the scientific community. Congratulations for the effort to study this important biome for humanity, although forgotten and highly degraded around the world

Reviewer 3 ·

Basic reporting

Generally, good writing and structure. Some sections require major clarification (see General comments for the author). Data is provided and the article is self contained.

Experimental design

Well defined aim and experimental design within the scope of the journal. Some parts of the methods require clarification (see General comments for the author).

Validity of the findings

The overall conclusions are well stated and not overly speculative. The novel case study provides seemingly robust findings, although some aspects require further evaluation (see General comments for the author).

Additional comments

The comments I provide here are not general but mostly very specific.

The manuscript provides a compelling case study for investigating the effects of restoration efforts on the soil microbiome in arid/semi-arid regions. The study investigates differences in physicochemical properties and the soil microbial community composition between intact, restored and degraded succulent thicket.
The study provides evidence of systematic shifts in bacterial community composition that could enable the detection of plant-associated indicator taxa. Furthermore, the restored regions and their microbiome seem to lie between the intact and degraded regions, suggesting a gradual transition.
Overall, the paper is well written, concise and easy to follow with a clear aim and experimental design. However, there are some aspects regarding methodology and interpretation of results that require clarification (see below). I can thus recommend publication after major revision.

1. It is not entirely clear how the ASV data was treated in the analysis. Was the data normalized? How? I think the differences in read numbers should be accounted for when comparing proportions across samples (1:10’000 reads is very different from 1:1’000’000). At least it should be reported or the authors could demonstrate that it does not affect their conclusions (e.g., regarding differences in diversity).
2. Several thresholds are used based on relative abundance. Again, this is difficult to interpret (see comment 1.). Also, I think the authors should show how different thresholds affect the outcome of the analysis.
4. I am not sure about the section on co-occurrence and the interpretation of interactions. It appears a bit out of the blue and does not substantially add to the conclusions of the manuscript. Maybe I missed something?

Comments as I went through the text (major comments marked with '*':
- L21: Maybe better state the years after replanting (instead of the year), since readers do not know when the study/sampling took place.
- L26: remove 'in'
- L27: "wide variety of habitats" does this refer to global (i.e. across continents and biomes: desert, forests,...) or regional habitats?
- L27-30: I suggest to split into two sentences and slightly adapt. "[...] threatened biome. It provides [...] function, concerning the restoration of degraded [...]"
- L50: Maybe break the sentence: "[...] 2013). These regions are referred to as [...]"
- L52: "[...] attributed to [...]"
- L77: please specify: "[...] the {region/ecosystem/soil?} remains [...]"
- L78-L82: The sentence could be written more clearly by first mentioning the attempts made ("replanting") and then (in a second sentence) detailing their success.
- L83: it is not clear what "these" refers to. I suggest to either specify or drop the word.
- L91-94: Here, I think also climate should be mentioned. Maybe this section could be expanded by providing some background for arid environments - e.g.: How is the soil/rhizosphere microbiome affected by water limitation? Relations of precipitation with pH, and organic matter contents?
- L98-100: Nice and clear!
- *L154-155: Why was this chosen? How about the other normalization methods (e.g., rarefy) that account for sampling depths?
- Fig. 1: The figure gives a good impression of the biome and nice overview of the sampled locations. Maybe add whitespace between the images. Also, the order is difficult to follow; maybe change to: (A) overview, (B + C) etc.
- Table S1: Please provide units for the measured values.
- *Fig. 2: The text is very small making the figure difficult to read (also some other figures). What was the proportion of variance explained in A?
- *L185-186: How was this tested? This should be clearly mentioned in the main text. Please, provide mean and standard deviation (also elsewhere in the text).
- L188: What "complex"?
- *L197-206: The significance test should be mentioned. Also, why were minimum/maximum values chosen for significance testing? This should be more explicit in the main text.
- L215: I think this suspicion could be tested by obtaining rainfall data (e.g., from satellite data https://disc.gsfc.nasa.gov/datasets).
- Fig. S5: It seems that the variability of diversity between samples was larger for the intact site compared to the degraded one. Maybe this is worth mentioning (also in context of the statistical tests used). Furthermore, if this pattern is consistent with the physicochemical measurements, it could indicate heterogeneity in the samples with presence of vegetation.

- L237: This is difficult to interpret. Was the data normalized (e.g., rarified)? Sequence reads might be high for a single sample with great sampling depth and could inflate the dominance of a particular species across samples.
- L237-238: "[...] three sampled zones [...]"
- *L235-246: It is not entirely clear to me how this was applied. Why was 95% of samples chosen? Others could be tested. Are those samples from the same group (e.g., 95% of intact samples)? I would expect that all samples are normalized (taking into account differing number of reads) and then the number of nonzero entries for each species is divided by the total number of samples to obtain the prevalence/ubiquity/occupancy.
- L251-252: How about the variation of these "rare" ASV? How did their proportion vary?
- *L268-270: It is a bit unclear what was used. E.g., are the variables soil density, soil moisture and temperature considered a "soil chemistry variable"? Also, maybe use "soil physicochemical property". Lastly, I could not follow where the 50% come from.

- *L276-279: Again, why were those thresholds chosen? Others should be tested. E.g., the authors could show how the number of ASV changes for differing thresholds.
- L288: Average pH value?

- *L303-308: I am not sure what this means in the context of network complexity. What metrics were compared to quantify the complexity (e.g., modularity, centrality, degree distribution)? How was the network constructed and what clustering method was used?
- *L309-315: The interpretation of interactions obtained from co-occurrences should be handled with care, especially in heterogeneous environments such as soil (see https://doi.org/10.1038/s41396-019-0463-3). This should be discussed.

- L338: I like this section for comparisons with the restoration. The intermediate position supports the robustness of findings.

---

## Round 0.2 · Minor Revisions

There are still issues to address. I do not expect them to give you much trouble.

Reviewer 1 ·

Basic reporting

no comment

Experimental design

no comment

Validity of the findings

no comment

Additional comments

no comment

·

Basic reporting

132 / 5000
Resultados de tradução
All inquiries have been answered. Although the reference requested to be citata was not inserted in the final text, unfortunately

Experimental design

All done

Validity of the findings

All done

Additional comments

All done

Reviewer 3 ·

Basic reporting

OK. Hypotheses could be more clearly stated.

Experimental design

OK. Methods need further writing.

Validity of the findings

OK

Additional comments

Overall, the manuscript improved and reads more fluently. However, there are still two issues that need to be resolved before I could recommend publication:
(i) The methods and the motivation behind them need to be clearly reported in the main text. Referring to computer code is not acceptable. It is not clear how data was treated. How exactly were temperature and humidity aggregated? Were proportions used or rarefied counts?
(ii) Different thresholds should be tested (as mentioned in my previous report) to show robustness of the findings. Commonly used thresholds are still arbitrary and thus not necessarily relevant. Showing how the outcome varies with different thresholds provides valuable insight into the robustness of analysis and distribution of the data (I am not suggesting to use multiple thresholds – only testing them).
Furthermore, some editing and proofreading is required.
Additional comments as I went through the text:
L88-90 delete 'elsewhere in the world'. What is meant here by ‘soil stability’?
Also, this hypothesis is always true and cannot be tested (i.e., they could always play a role). Change to 'Soil bacterial diversity is important for maintaining ...' and then introduce your own hypothesis: 'We hypothesize that increased diversity increases soil ... in the Albany Subtropical Thicket.', however, given the aim of the study (L103) it could be 'We hypothesize that changes in environmental factors due to the loss of succulent thicket vegetation explain shifts in soil microbiome composition.' (Ideally a hypothesis would also come with defined direction)
L96 The part on the 'feedback loop' is not relevant for this study and can be removed (including the reference Rocca et al. 2019 that introduced a database and no mechanism).
L116 delete 'as part of a restoration experiment'
L121 change '... and a "site" is the point ...'
L122 change '... soil chemical analysis ...'
L128-131 This sentence is unclear. First introduce the RDA and the motivation for using it. Then in a second sentence explain how the variables were aggregated (not 'generated' I hope...). What was the input? Mean and std? Why? Also, later there is mention of daily values. This needs to be cleared up. How were things aggregated? The authors refer to their code in the point by point response, but I do not think this is ok. Most readers do not have the time to read 1000 lines of code. Additionally, GitHub is not a proper archive and something like Zenodo (with DOI) should be used to guarantee future access.
L128 Why was such a high resolution chosen? Were the values filtered/smoothed? Was the raw data used to do the analysis? e.g., a single temperature/humidity peak within two hours could have been chosen as the maximum value used for further analysis (and would be considered representative for the sample)? Is this robust?
L162-164 Does this make sense? I suggest the authors repeat the alpha diversity analysis on normalized (rarefied) data. Otherwise I think it is neither representative nor interpretable. How big was the variation in reads (not written in the main text - 3x is not nothing and depends on the range e.g., 10000-30000)? How does the (expected) number of individuals (cells) compare to the number of reads?
L166 'agglomerated'?
L170 'The final model was tested and evaluated by ANOVA to ensure that all terms were statistically significant' what is the point of this (statistically speaking)? If a term is insignificant it should be kept in the model and it would not affect the outcome.
L176 please rephrase. Is it 'different in abundance' or 'differentially abundant'?
L182 replace '2' with 'two'
L188 what is 'shallow-subsurface'?
L201 How about the water retention due to organic matter?
L228 move 'content' before the bracket after 'NH4-N'
L299 'These results replicate previous findings for ...'
L246-248 Please rewrite. This sentence is too long and very difficult to follow. E.g., if everything is consistently aggregated to daily values (and reported in the methods) then it would not need to be stated again. Why should the maximum and minimum be compared here?
L253-254 Was this 'lens' observed at the same site? If not, can it be generalized? I suggest to be clear that this is speculation. In general organic matter would decrease bulk density and increase porosity and, thus, water infiltration.
L264 This would not be 'unexpected' if dispersal is not limiting (due to proximity of sites),; i.e., richness is constant and only abundance changes.
L283 The dominance of few species is well known in ecology and does not only apply to soil microorganisms.
L313 This suggests that the water balance is altered (e.g., http://www.nature.com/doifinder/10.1038/nature20139) and could affect the community structure (i.e., evenness). I suggest to reconsider interpretation (L316-318) since both components (humidity and temperature) are responsible for changes in water balance. Also please fix the typos and introduce the hypothesis at the beginning.
L399 remove 'tend to'

---

## Round 0.3 · accepted · Accept

Thank you for addressing the final issues.